# Thin Layers of SrTiO$_3$-TiO$_2$ with Eutectic Composition for Photoelectrochemical Water Splitting

Jaroslaw Sar [1,2,*], Katarzyna Kołodziejak [1,2], Krzysztof Orliński [1], Michal Gajewski [1], Marian Teodorczyk [1] and Dorota Anna Pawlak [1,2]

[1]  Łukasiewicz Research Network—Institute of Microelectronics and Photonics, Al. Lotników 32/46, 02-668 Warsaw, Poland
[2]  Centre of Excellence ENSEMBLE3 Sp. z o. o., Wolczynska Str. 133, 01-919 Warsaw, Poland
*  Correspondence: sar.jaroslaw@gmail.com

**Abstract:** Hydrogen as a potential fuel of the future can be produced in a photoelectrochemial water splitting process. Herein, we demonstrate the fabrication of photoelectrochemical electrodes based on SrTiO$_3$-TiO$_2$ with a eutectic composition on titanium and conductive glass FTO (fluorine doped tin oxide) substrates by magnetron sputtering. The XRD and SEM/EDS reveal the amorphous nature and homogeneity of the sputtered material. The influence of the layer thickness on the photoactivity was investigated. There were three-layer thicknesses (50, 350, and 750 nm) selected by sputtering for 12, 80 and 220 min for the preparation of photoelectrodes. The photoelectrochemical analysis confirms the photoactivity of the obtained layers under illumination with a xenon lamp (600 mW/cm$^2$). The highest photocurrent density of $11.8 \times 10^{-3}$ mAcm$^{-2}$ at 1.2 V vs. SCE was recorded for a layer thickness of 50 nm on titanium as better matching between the Ti work function and the conduction band.

**Keywords:** magnetron sputtering; SrTiO$_3$; TiO$_2$; PEC; water splitting; hydrogen

## 1. Introduction

The alternative to hydrogen production through conventional water electrolysis is through photoelectrochemical (PEC) cells. PEC water splitting implies the junction of semiconductors and liquid, where the sunlight is captured, and water molecules are directly electrolyzed producing gaseous hydrogen and oxygen. PEC cells are built from photoactive semiconductor material, n-type for the photoanode and p-type for the photocathode. Photons with energy equal to or higher than the band gap generate electron–hole pairs in the semiconductor. Electrons travel from the photoanode to the photocathode where they reduce H$^+$ into H$_2$. At the same time, the holes react with water molecules producing O$_2$. From the large number of semiconductors, only a few fulfill the necessary criteria for efficient separation of a photogenerated electron–hole, photoresponse matched to the solar spectrum and stability under harsh operating conditions [1]. The potential required for thermodynamic water splitting is 1.23 V, but taking into account the energy losses, the optimum band gap of the semiconductor should be around 2 eV [2]. Moreover, the position of the conduction band should be more negative than the hydrogen evolution potential, while the valence band edges should be more positive than the oxygen evolution potential [3]. None of the semiconductors can satisfy the criteria of stability, band gap and position of band gap edges. Materials that meet the above criteria are eutectic composites. Eutectic composites are characterized by the formation of two or more non-mixable crystals from a completely mixable melt, which leads to the formation of self-organized micro/nanostructures [4]. Directionally solidified eutectics (DSE) may be defined as composite materials with a complex and homogeneous micro- or nanostructure, which controls their properties [5]. Eutectic composites are broadly investigated in terms of special geometrical motifs (lamellar, fibrous, globular and spiral) [6,7], the plasmonic effect [8,9], electromagnetic properties [10] and,

recently, photoelectrochemistry [11–13]. Eutectic composites are able to compete with other materials because of their (i) high crystallinity (as single crystals) [14], (ii) sharp interfaces between the component phases [15], (iii) potential use in designing materials with various component materials, enabling broadband absorption (UV-Vis) and (iv) utilization phases, which are hard to achieve or are unavailable in another way. The aim of this work was to fabricate the photoelectrochemically active layers of a $SrTiO_3$-$TiO_2$ composite with a eutectic composition by magnetron sputtering. Both components of the $SrTiO_3$-$TiO_2$ eutectic system have a wide bag gap corresponding to the absorption of UV light [16,17], and their valence and conduction bands overlap the water redox potentials [18]. The layers were deposited on titanium and glass with fluorine doped tin oxide (FTO) with a controllable sputtering time in order to investigate the effect of layer thickness on the photoelectrochemical performance. The deposition of $SrTiO_3$ and $TiO_2$ is a common subject in areas such as computer memories, sensors, IR detectors, catalysts and others. Researchers use the advantages of magnetron sputtering to modify and improve the properties of materials. It can be found in the literature that magnetron sputtering was used for uniform doping to get Al-$SrTiO_3$ [19] and Nb-$TiO_2$ [20]; for the sandwich structure of $SrTiO_3$/$TiO_2$/$SrTiO_3$ [21]; for $SrTiO_3$-$TiO_2$ heterostructures [22–24]; and for the deposition of $TiO_2$ on glass, Pt, Si [25] and $SrTiO_3$ thin films, for the application in solar energy harvesting [26]. In this work, we present the simultaneous sputtering of $SrTiO_3$ and $TiO_2$ in order to form a homogeneous distribution of both components on Ti and FTO substrates. It is possible because of the preparation of the $SrTiO_3$-$TiO_2$ sputtering target with the eutectic composition and use for photoelectrochemical water splitting.

## 2. Materials and Methods

### 2.1. Magnetron Sputtering

Thin layers of $SrTiO_3$-$TiO_2$ eutectic composites were deposited by magnetron sputtering. The ceramic target for sputtering was obtained by mixing SrO and $TiO_2$ powders in a ratio corresponding to a eutectic composition of 23 mol. % of SrO and 77 mol. % of $TiO_2$ [27]. The mixture of powders was preheated at 70 °C and then synthesized at 1100 °C. The prepared material was grated with a 10% water solution of PVA to form granules. The ceramic target in a disk shape with a diameter of 100 mm and thickness of 5 mm was pressed and sintered at 1300 °C for 12 h in air. Magnetron sputtering was performed in a ULVAC vacuum sputtering machine. Before the sputtering, the substrate was cleaned by successive sonication in water and ethanol for 10 min each. The distance between the target and the substrate was set at 43 mm. The substrate temperature was ambient before the sputtering, and it was not heated or cooled down during the process. Its temperature changed to around 60 °C while heating from the generated plasma. The relationship between the sputtering time and the layer thickness was investigated.

### 2.2. Material Characterization

The thickness of the deposited layer was analyzed by a Veeco Dektak 150 surface profilometer. During the initial deposition, the substrate surface was partially covered in order to create the gap between the bare substrate and the deposited material. The X-ray powder diffraction analysis was performed on the as-grown samples using a Siemens D500 powder diffractometer equipped with a semiconductor Si:Li detector from 20–120° with the θ/2θ scanning mode with a step size of 0.02° and a counting time of 10 s/step, using Cu Kα radiation (1.5406 Å). Phases were identified using the International Centre for Diffraction Data PDF4+ database [28]. Additionally, the deposited layers were analyzed by grazing incidence X-ray diffraction (GIXD). SEM images were performed on the as-deposited samples using a Cross Beam Workstation (Carl Zeiss SMT, Oberkochen, Germany) combined with an energy dispersive X-ray spectrometer.

### 2.3. Photoelectrochemical (PEC) Analysis

The photoelectrochemical analysis was performed in a Teflon/PTFE cell with a quartz window. Measurements used a three-electrode configuration: $SrTiO_3$-$TiO_2$ electrodes served as a working electrode (WE), a saturated calomel electrode (SCE) served as the reference electrode (RE) and a platinum electrode as the counter electrode (CE). A mixture of $H_2SO_4$ and $Na_2SO_4$ with a pH of 2 was used as an electrolyte. All PEC was performed in CHI-660D potentiostat using open-circuit potential (OCP) mode versus time and cyclic voltammetry in the potential range of $-0.3$–1.7 V and a 10 mV/s scan rate. Electrochemical impedance spectroscopy (EIS) was carried out at 1 V vs. SCE potential in the frequency range from 1 MHz to 0.01 Hz. All techniques were used with illumination and in dark conditions. A 150 W xenon lamp from solar light with 600 mA/cm$^2$ irradiation calibrated by a Solar Light PMA 2144 Class II pyranometer was used as a light source.

## 3. Results and Discussion

### 3.1. Layers of SrTiO$_3$-TiO$_2$-Eutectic Composition

The $SrTiO_3$-$TiO_2$ layers were deposited by sputtering on two types of substrates: titanium and glass with fluorine doped tin oxide (FTO). These depositions were initially performed to define the relationship between sputtering time and layer thickness. The deposition parameters, such as $P_{Ar}$ vacuum level expressed in argon pressure, power of the device causing the voltage to be applied between the substrate and the target, target to substrate distance, sputtering time and the measured thickness of the layer are presented in Table 1. The layer thickness increases logarithmically with increasing sputtering time (Figure 1a). Based on the initial depositions, there were three deposition settings selected to obtain layer thicknesses around 50, 350 and 750 nm (Figure 1b–d). The high resolution of the profilometer also allowed for the determination of measuring point inaccuracies related to roughness and/or unwanted contamination of the substrate, as well as the layer.

**Table 1.** List of the deposition parameters: $P_{Ar}$ vacuum level expressed in argon pressure, power of the device, target to substrate distance, sputtering time and the measured thickness of the layer.

| $P_{Ar}$ (hPa) | Power (W) | Target to Substrate Distance (mm) | Sputtering Time (min) | Thickness (nm) |
|---|---|---|---|---|
| $2 \times 10^{-2}$ | 400 | 43 | 12 | 40–60 |
| $2 \times 10^{-2}$ | 400 | 43 | 60 | 220–250 |
| $2 \times 10^{-2}$ | 400 | 43 | 80 | 330–360 |
| $2 \times 10^{-2}$ | 400 | 43 | 100 | 450–480 |
| $2 \times 10^{-2}$ | 400 | 43 | 150 | 580–620 |
| $2 \times 10^{-2}$ | 400 | 43 | 220 | 720–770 |

### 3.2. Material Characterization

Figure 2a shows the XRD patterns of the target prepared for sputtering. There were distinguished sharp peaks that can be attributed to the rutile ($TiO_2$) phase and tausonite ($SrTiO_3$) [20]. The sintering at 1300 °C/12 h in air allowed the synthesis of desired composition phases in the target. However, XRD patterns of $SrTiO_3$-$TiO_2$ layers deposited on the Ti substrate did not show corresponding peaks for constituent phases (Figure 2b). No evidence of the crystalline structure can indicate the amorphous nature of the deposited layers. The only recorded peaks were attributed to the Ti substrate. Additionally, the GIXD patterns (the inset of Figure 2b) show only weak reflection peaks attributed to the substrate. The rest indicates amorphous background. Similar results were registered for all deposited layer thicknesses regardless of the used substrate.

The SEM images presented in Figure 3 reveal the microstructure of the materials. In all investigated samples, the microstructure was homogeneous without any abnormal precipitates. In high-magnification pictures, it can be observed that with increasing sputtering time, the sharp edges between grains become smoother. The microstructure did not show

the existence of split-ring resonators typical structures that like to form during the pulling of the $SrTiO_3$-$TiO_2$ eutectic crystals [7,13,29].

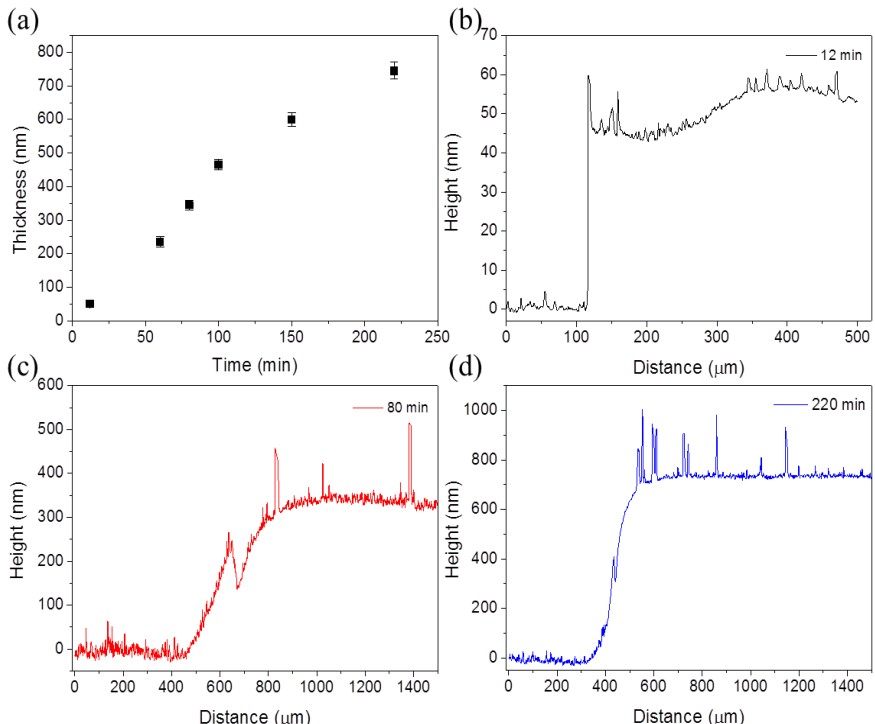

**Figure 1.** Thickness of the layers measured by profilometry: (**a**) relationship between thickness of the layer and sputtering time; height of the layer measured after (**b**) 12 min, (**c**) 80 min and (**d**) 220 min of sputtering.

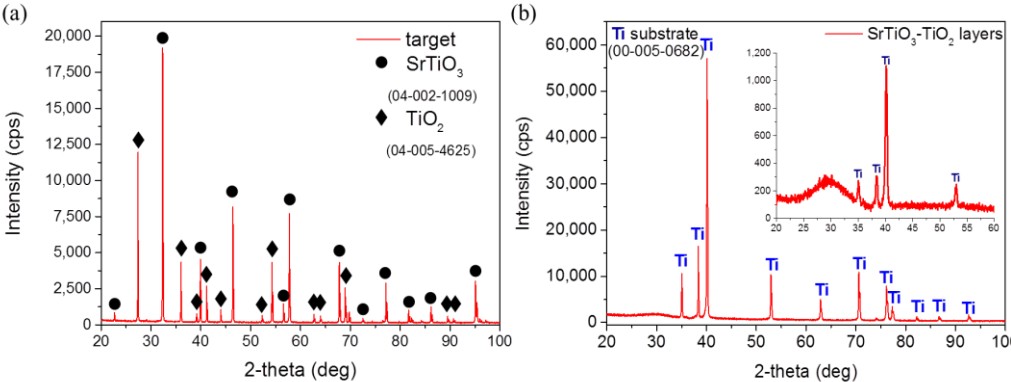

**Figure 2.** X-ray diffraction of (**a**) prepared target for sputtering revealing $SrTiO_3$ and $TiO_2$ phases and (**b**) $SrTiO_3$-$TiO_2$ layers deposited on Ti substrate with the 220 min long sputtering process. The inset represents GIXD patterns focused on 2-theta range of 20–60 deg. The XRD card numbers were put in the brackets for $SrTiO_3$, $TiO_2$ and Ti.

Good adhesion between FTO–glass substrate and deposited $SrTiO_3$-$TiO_2$ was confirmed for both 80- and 220-min sputtering times (Figure 4). In the case of the 12 min long sputtering, the recognition of a 50 nm thick layer was difficult.

The chemical composition of the obtained composite was examined by SEM equipped with an energy dispersive X-ray spectrometer (EDS) and presented in the form of maps for individual elements (Figure 5). The distribution of the key elements (Sr, Ti, and O) constituting the composite is homogeneous in the studied area. The presence of foreign inclusions was not registered.

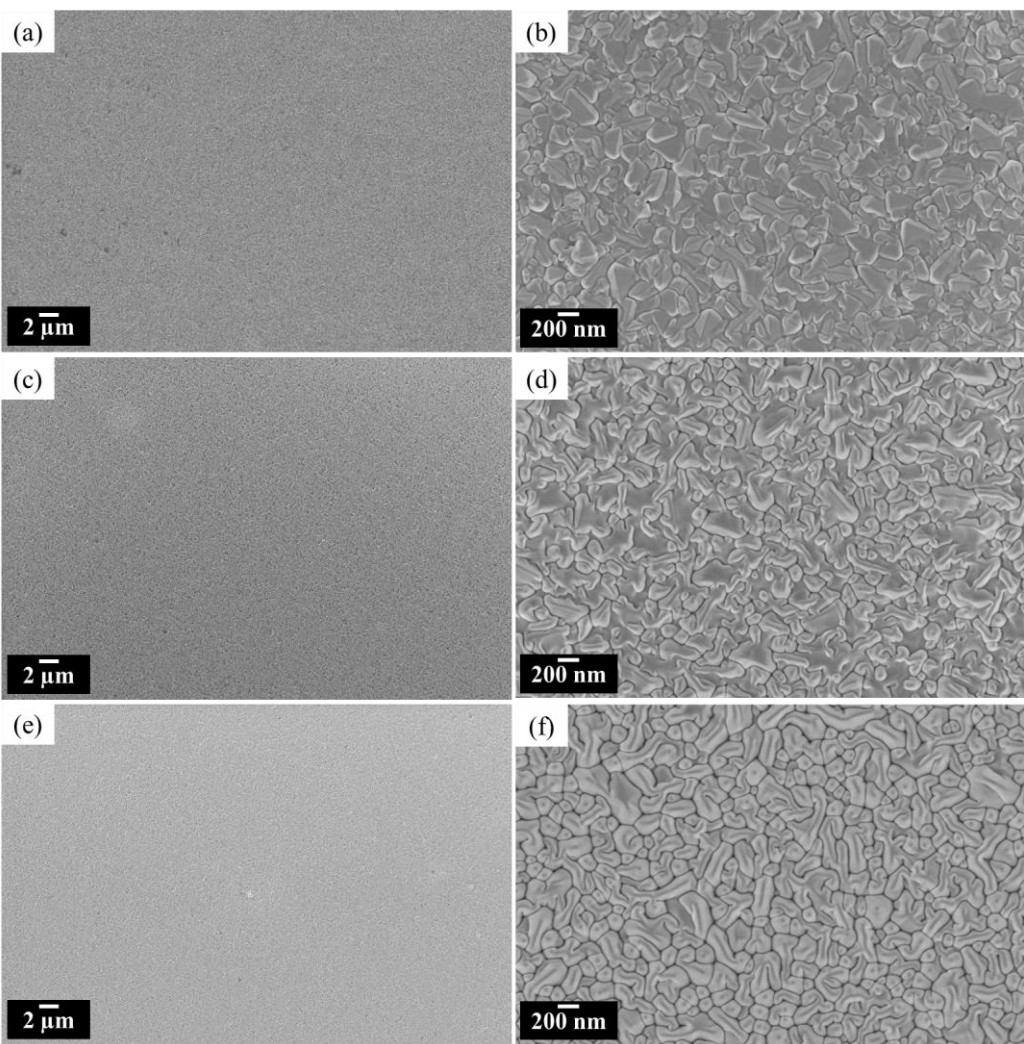

**Figure 3.** SEM images of the SrTiO$_3$-TiO$_2$ layers after sputtering for (**a**,**b**) 12 min; (**c**,**d**) 80 min; and (**e**,**f**) 220 min.

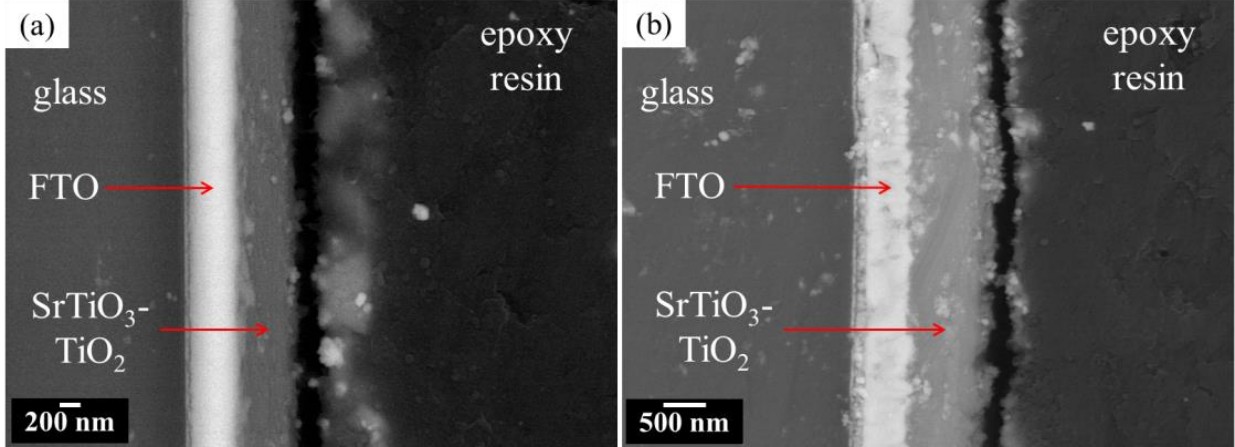

**Figure 4.** SEM picture made on cross-section of SrTiO$_3$-TiO$_2$ after sputtering for (**a**) 80 min and (**b**) 220 min.

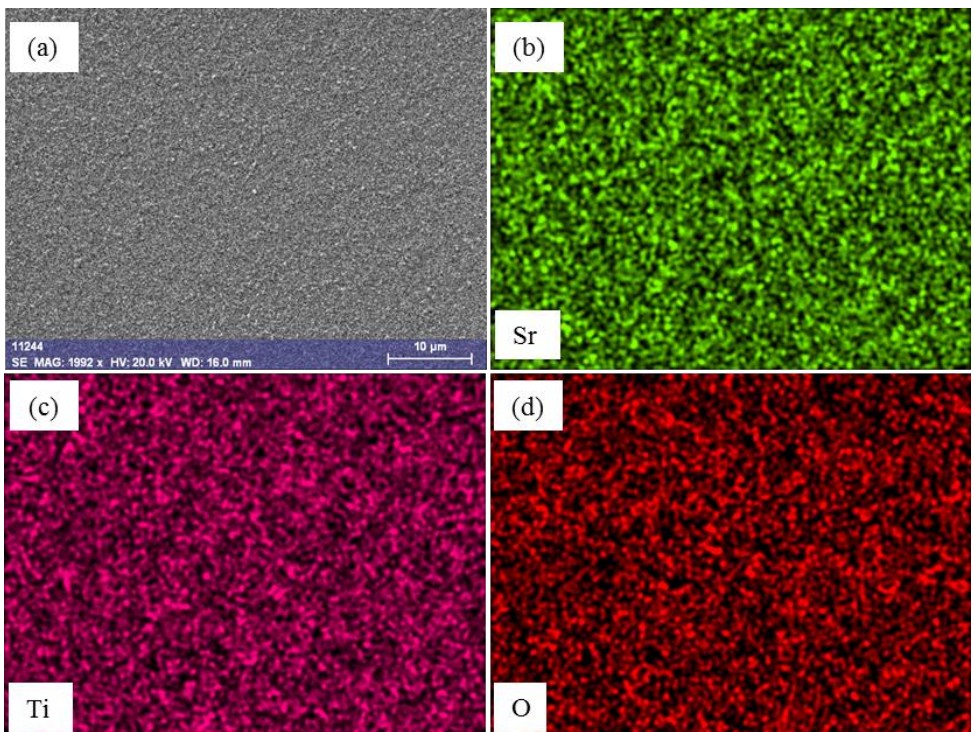

**Figure 5.** SEM photo with EDS analysis taken for 350 nm $SrTiO_3$-$TiO_2$ at FTO: (**a**) SEM photo at 1992× magnification; EDS mapping of (**b**) Sr, (**c**) Ti and (**d**) O elements.

### 3.3. Photoelectrochemical (PEC) Measurements

PEC measurements were started with the registration of the open-circuit potential (OCP) over time when the light source is cyclically turned on and off (light on/off). Figure 6 shows that OCP for 50 nm thick layers on the FTO substrate has changed as a result of switching the light source on/off, indicating the photosensitivity of the tested material. The original OCP value of −13 mV is decreased to −192 mV under the light and then the original OCP is gradually recovered when the light is off. The decrease in OCP describes the n-type semiconductivity of the electrodes, which was registered for all tested electrodes. In the n-type semiconductors, the photogenerated holes flow to the surface-electrolyte interface where the oxygen evolution reaction takes place. The OCP shifts are in accordance with previous findings [4,30,31].

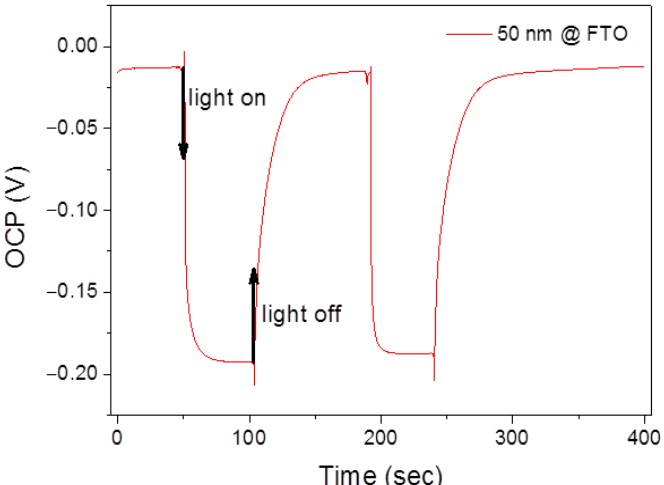

**Figure 6.** Changes in the open-circuit potentials over time when turning the light source on and off.

The photoelectrochemical analysis was continued through the current–voltage characteristics under alternating dark and illuminated conditions (so-called "chopped mode"). Despite the substrate, the lowest photocurrent densities were recorded for the thinnest layers. Figure 7 shows the effect of layer thickness on the photocurrent densities. An improvement in photoresponse was observed along with the decreasing thickness of the photoactive layer. The photocurrent densities were $11.8 \times 10^{-3}$, $1.82 \times 10^{-3}$ and $0.37 \times 10^{-3}$ mAcm$^{-2}$, successively for 50, 350 and 750 nm, at the potential of 1.2 V vs. SCE for layers deposited on Ti. This improvement can be explained by the relatively small grains: the photogenerated holes have enough time to diffuse to the interface with the electrolyte and to oxidize the water [12]. Additionally, the incident light is scattered at the grain boundaries, leading to enhanced optical absorption [32].

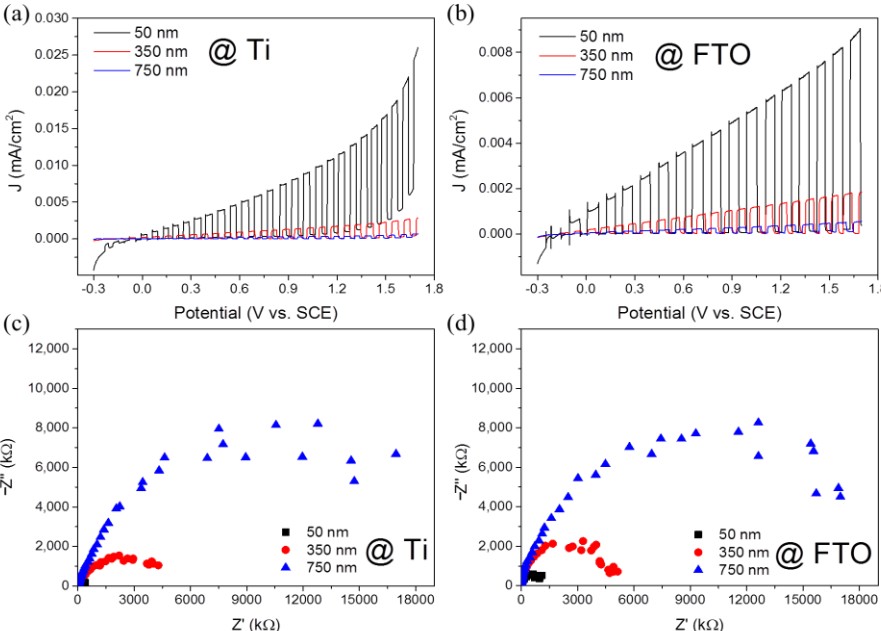

**Figure 7.** The influence of layer thickness on the obtained photocurrent densities presented in chopped current–voltage curves for layers deposited on (**a**) Ti substrate and (**b**) FTO substrate. The influence of layer thickness under illumination on electrochemical impedance spectroscopy is presented in Nyquist plots for layers deposited on (**c**) Ti substrate and (**d**) FTO substrate.

At the same time, the photocurrent densities recorded for electrodes on FTO substrates were lower at each deposition thickness in comparison with Ti ones. The differences between the observed photoresponses come from differences in the potential of the ohmic contact [11]. The conduction band levels for TiO$_2$ and SrTiO$_3$ are $-4.21$ and $-4.24$ eV, respectively [33], whereas the work function of polycrystalline titanium is 4.33 eV. The commonly cited value for work function for FTO is 4.40 eV [34]; however, as described by Helander et al. [35], it can be as high as 5.00 eV, as is found for commercial FTO. The higher mismatch between the conduction band and the work function can lead to the low photoresponse observed for FTO-based photoanodes.

The highest obtained photocurrent density of $11.8 \times 10^{-3}$ mAcm$^{-2}$ for 50 nm thick layers on Ti substrate was smaller than the reported 8.5 mAcm$^{-2}$ for SrTiO$_3$-TiO$_2$ eutectic-based photoelectrodes [11]. The photoelectrochemical performance of our electrode is limited because of a lack of crystalline contact within the grain and between both phases. The amorphous nature of deposited layers increases recombination losses [36], diminishes charge-carrier transport [14] and worsens band gap and Fermi-level positions [37].

The investigation of changes in charge transfer resistance in the electrochemical impedance spectroscopy was carried out in dark and light conditions. The experimental data for EIS spectra presented in the Nyquist plots (Figure 7b,c) were fitted to the

equivalent-circuit model, composed of two RC links [38]. The first RC link corresponding to the high-frequency region describes the resistance (Rsc) and capacitance (Csc) of the space charge zone. The second RC link at the low-frequency region describes charge transfer resistance (Rct) and the electric double-layer capacitance (Qdl). The fitting results are listed in Table 2. The photosensitivity of $SrTiO_3$-$TiO_2$ layers observed during OCP was confirmed by a strong decrease in Rct under illumination, regardless of the material thickness and substrate. Moreover, Rct was decreased with the thickness reaching the lowest value of 0.41 M$\Omega$ for 50 nm thick layers deposited on Ti. The comparison of substrates showed lower charge transfer resistances at each thickness for layers sputtered on the Ti substrate. EIS results are consistent with results from cyclic voltammetry.

**Table 2.** The fitting results of the experimental data for the proposed equivalent-circuit model for $SrTiO_3$-$TiO_2$ layers sputtered on Ti substrate and FTO substrate under dark and light conditions: where Rs is series resistance, Rsc is space charge resistance, Csc is space charge capacitance, Rct is charge transfer resistance, Qdl is electrode capacitance in constant phase element, and n is constant phase exponent.

| $SrTiO_3$-$TiO_2$ @ FTO | | | | | | |
|---|---|---|---|---|---|---|
| Thickness (nm) | | Rs ($\Omega$) | Rsc ($\Omega$) | Csc (F) | Rct (M$\Omega$) | Qdl (F s(n − 1)) | n |
| 750 | Dark | 30.0 | 75.3 | $3.51 \times 10^{-9}$ | 23.2 | $1.21 \times 10^{-8}$ | 0.95 |
| 750 | Light | 29.8 | 74.4 | $3.57 \times 10^{-9}$ | 10.5 | $1.23 \times 10^{-8}$ | 0.95 |
| 350 | Dark | 23.5 | 84.5 | $3.88 \times 10^{-9}$ | 53.1 | $1.94 \times 10^{-8}$ | 0.98 |
| 350 | Light | 23.0 | 82.9 | $3.85 \times 10^{-9}$ | 4.29 | $2.05 \times 10^{-8}$ | 0.98 |
| 50 | Dark | 28.9 | 99.8 | $3.48 \times 10^{-9}$ | 11.0 | $2.87 \times 10^{-7}$ | 0.96 |
| 50 | Light | 22.0 | 102.0 | $3.01 \times 10^{-9}$ | 1.60 | $4.71 \times 10^{-7}$ | 0.91 |

## 4. Conclusions

The thin layers of $SrTiO_3$-$TiO_2$ composite material with a eutectic composition were successfully deposited on titanium and FTO substrates by sputtering. We were able to control the thickness of deposited layers by changing the sputtering parameters, such as the vacuum level, the voltage applied between the substrate and the target and the sputtering time. The deposited materials were found to be amorphous and homogeneous, both in terms of microstructure and composition. Material thicknesses of 50, 350 and 750 nm were achieved by carrying out the sputtering for 12, 80 and 220 min, respectively. These three-layer thicknesses were selected for the photoelectrochemical studies (OCP and chopped-CV). All prepared electrodes showed photoelectrochemical activity under a 150 W Xe-arc lamp with an irradiance intensity of 600 mWcm$^{-2}$. The photocurrent densities were increasing while the thickness of the active layer was lowering, and the photocurrent density reached the highest value of $11.8 \times 10^{-3}$ mAcm$^{-2}$ at 1.2 V vs. SCE for a layer thickness of 50 nm. The highest values for photocurrents were recorded for $SrTiO_3$-$TiO_2$ sputtered on titanium substrates as a result of better matching of the work function with the conduction band. Further improvement in deposited layers is expected to lie in providing the full potential of eutectic composites by their high crystallinity, which allows matching the band gaps and the positions of band edges of constituent components and improving the photogenerated carriers transport ability. Thin layers of eutectic composites open up several unique opportunities: (a) manufacturing of hybrid materials, including complex oxides and oxide-non-oxide composite semiconductors with improved photoelectrochemical performance, (b) a better understanding of the behavior of highly crystalline semiconductor hetero-junctions formed in nanoscale multicomponent systems, and (c) a significant impact on areas beyond photoelectrochemistry such as nanotechnology, sensing, chemistry and photovoltaics.

**Author Contributions:** Conceptualization, methodology, investigation, project administration, and writing—original draft preparation, J.S.; methodology, investigation, writing—original draft preparation, K.K.; methodology, investigation, K.O.; methodology and investigation, M.G.; methodology, M.T.; conceptualization, methodology, writing—review and editing and supervision D.A.P. All authors have read and agreed to the published version of the manuscript.

**Funding:** The research has been supported by a SONATA Project (2016/23/D/ST5/02882) from the National Science Centre. The authors thank the ENSEMBLE3 Project carried within the Teaming for Excellence Horizon 2020 programme of the European Commission (GA No. 857543), and the International Research Agendas Programme (MAB/2020/14) of the Foundation for Polish Science co-financed by the European Union under the European Regional Development Fund and Teaming Horizon 2020 programme of the European Commission.

**Institutional Review Board Statement:** Not applicable.

**Informed Consent Statement:** Not applicable.

**Data Availability Statement:** Not applicable.

**Conflicts of Interest:** The authors declare no conflict of interest.

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
