# Peer review of "Thin Layers of SrTiO3-TiO2 with Eutectic Composition for Photoelectrochemical Water Splitting"

_coatings, doi:10.3390/coatings12121876_

Round 1

Reviewer 1 Report

Paper entitled ":Thin Layers of SrTiO3-TiO2 with Eutectic Composition for Photoelectrochemical Water Splitting" prezsented some important results.

Suggest minor revision with following comments

Magnetron sputtering: Please include the sputtering condition: Sputtering power: power density, pressure target to substrate distance, pre-cleaning, substrate temperature etc. in the text. Also, mention the table

Please mention the wavelength of Cu Kα radiation.

Was standard test done for the Photoelectrochemical (PEC) analysis?

Does the shape of the curve in Fig.5 tell; something?

Otherwise, paper is ready to go.

Reviewer 2 Report

The authors reported the fabrication of photoelectrochemical electrodes based on SrTiO3-TiO2 with eutectic composition on titanium and conductive glass FTO (Fluorine doped Tin Oxide) substrates by the magnetron sputtering. The work is not innovative enough, and the data is not sufficient, I suggest re-evaluation after major revisions.

1. The highest obtained photocurrent density of 11.8 x 10−3 mAcm−2 in this work was far less than 8.5 mAcm−2 reported for SrTiO3-TiO2 eutectic based photoelectrode (https://doi.org/10.1016/j.apcatb.2017.01.054). What is the innovation of this article?

2. Please provide the XRD card number of each phase in Figure 2.

3. Why can't the peaks of SrTiO3 and TiO2 be measured after SrTiO3-TiO2 layers are deposited into Ti substrate? According to the XRD peak strength, this should not be the case. The sputtering products can be scraped off and then tested, or other evidence can be provided to confirm the crystal phase composition of the sputtered substance on the Ti substrate.

4. The cross-section of the film layer is the visual evidence to judge the quality of the film, and the authors need to provide it.

5. The Mott-Schottky plots and the electrochemical impedance spectroscopy analysis of the samples should be provided.

Reviewer 3 Report

This manuscript present on the Thin Layers of SrTiO3-TiO2 with Eutectic Composition for Photoelectrochemical Water Splitting. This work reported that SrTiO3-TiO2 eutectic composites were deposited by Magnetron Sputtering. In addition, the relation between the sputtering time and the layer thickness was investigated. The authors have not carried out systematic investigation. The results are not interesting by the experimental/analytical evidence. I have several questions regarding the fundamental aspects of the study: I do not find that this paper is enticing and my observations are listed as below:

- In fact, deposition of SrTiO3-TiO2 by using sputtering technology have been widely reported, and there have been many in-depth studies and discussions on the effects of varied sputtering process parameters on the properties of SrTiO3-TiO2. However, the authors did not review and discuss previous studies in the Introduction paragraph of the manuscript. Moreover, the experiments in this work were not well systematically designed and the results were not well presented.

- The aim of the study is so clear, please discuss/add details.

-Please connect the discussion with each other to obtain results.

-“On high magnification pictures, it can be observed that with increasing sputtering time the sharp edges between grains become smoother and the gran boundaries are more developed”. There are also several erroneous concepts, please check once again.

- Authors need to do some annealing processes and check their effect on the layers and application.

-The authors have used only XRD and SEM analysis techniques which is not enough to characterize the materials in detail.

- The authors need to justify the compositions with analysis like XPS.  For more clear understandings of the electronic structure, XPS spectra of Ti, O spectrum should be provided.

In addition, the authors only used cyclic voltammetry to test the performance of some samples.

- Overall, lack of novelty, missing information, not much convincible data and explanation, and overall content does not seem fit. 

Round 2

Reviewer 2 Report

  • The authors have revised all the issues, and I agree to accept this article.

Reviewer 3 Report

The authors have substantially improved the manuscript. Now can be considered for publication after a few minor revisions. 

The authors are suggested to improve the justification of current work in the introduction with the help of https://doi.org/10.1016/j.matchemphys.2022.126861, https://doi.org/10.1016/j.surfin.2022.102410, https://doi.org/10.1016/j.ceramint.2022.10.131.

-"Researchers use the advantages of the magnetron sputtering to modify and improve properties of materials" It is confusing.